# Patagonian Berries: Healthy Potential and the Path to Becoming Functional Foods

**DOI:** 10.3390/foods8080289

**Published:** 2019-07-26

**Authors:** Lida Fuentes, Carlos R. Figueroa, Monika Valdenegro, Raúl Vinet

**Affiliations:** 1Centro Regional de Estudios en Alimentos Saludables (CREAS), CONICYT-Regional GORE Valparaíso Proyecto R17A10001, Avenida Universidad 330, Placilla, Curauma, Valparaíso 2340000, Chile; 2Institute of Biological Sciences, Campus Talca, Universidad de Talca, Talca 3465548, Chile; 3Escuela de Agronomía, Facultad de Ciencias Agronómicas y de los Alimentos, Pontificia Universidad Católica de Valparaíso, Calle San Francisco s/n, Casilla 4-D, Quillota 2260000, Chile; 4Laboratorio de Farmacología, Centro de Micro Bioinnovación (CMBi), Facultad de Farmacia, Universidad de Valparaíso, Gran Bretaña 1093, Valparaíso 2360102, Chile

**Keywords:** maqui, murta, calafate, arrayán, Chilean strawberry, berries, functional foods

## Abstract

In recent years, there has been an increasing interest in studying food and its derived ingredients that can provide beneficial effects for human health. These studies are helping to understand the bases of the ancestral use of several natural products, including native fruits as functional foods. As a result, the polyphenol profile and the antioxidant capacity of the extracts obtained from different Patagonian native berries have been described. This review aims to provide valuable information regarding fruit quality, its particular compound profile, and the feasibility of producing functional foods for human consumption to prevent disorders such as metabolic syndrome and cardiovascular diseases. We also discuss attempts concerning the domestication of these species and generating knowledge that strengthens their potential as traditional fruits in the food market and as a natural heritage for future generations. Finally, additional efforts are still necessary to fully understand the potential beneficial effects of the consumption of these berries on human health, the application of suitable technology for postharvest improvement, and the generation of successfully processed foods derived from Patagonian berries.

## 1. Introduction

When we imagine a place like Patagonia, it is impossible not to evoke images of extraordinary beauty like southern ice fields. However, a walk through this place also allows us to contemplate ancestral traditions that include the use of many native species. This southernmost region of the South American continent extends from 37° S to Cape Horn, at 56° S, whose geography is characterized by the Andes range, which is both the continental watershed and the international limit between Argentina and Chile. It includes the Pacific and Atlantic coasts and lowlands, the southern archipelagos and tablelands, and the valleys and high plains extending between the Andes and the Atlantic Ocean [1].

The Andean temperate forests of Patagonia have a great diversity of plants with medicinal properties [2,3]. The use of medicinal and edible native plants is a long-standing tradition in the Mapuche communities of Southern Argentina and Chile [4,5,6]. An ethnobotanical survey conducted in rural villages of San Martin de Los Andes, Argentina, showed the use and knowledge of about 40 and 47 native plants, respectively [5]. Unfortunately, this ancient knowledge tends to disappear in the younger generations [5]. Moreover, the effects of human activity (e.g., an increase in dwelling number) and the invasion of alien plants can reduce the availability of forest-associated gathering sites. Therefore, the use of food derived from non-cultivated plants as part of the diet could be a tradition susceptible to disappearing [7,8,9] and the cultural, social, and economic aspects must be evaluated comprehensively if these traditions are to be maintained for future generations [8,9].

In recent years, the interest in food or ingredients that provide beneficial effects for human health has increased. As a result, many native fruits from different continents have been studied as a source of functional foods [9,10,11,12,13,14,15,16]. In Chilean Patagonia, edible fruits come from woody or shrub forest species belonging to the Elaecarpaceae, Berberidaceae, and mainly Myrtaceae families [15,16], and creeping plants belong to Rosaceae family. These species present fruits rich in antioxidant and functional compounds, such as *Aristotelia chilensis* (maqui), *Berberis microphylla* (calafate), *Ugni molinae* (murta), *Luma apiculata* (arrayán), and *Fragaria chiloensis* (Chilean strawberry), among others [15,16,17,18,19,20,21,22,23] (Table 1). In Chile, these native species are mainly distributed from the Coquimbo to Magallanes regions (Latitude 31° to 55°), with Chilean Patagonia being the common region for all fruits analyzed in the present review (Table 1).

Most of the traditional uses of these fruits include consumption as fresh and dried fruits or being used to make tea, jam, cakes, juice, alcoholic beverages, and textile tinctures. Moreover, they have tremendous functional potential due to their high antioxidant values, particularly flavonol and anthocyanin contents and promissory bioassay results as anti-inflammatory, antidiabetic, and hypolipidemic agents [11,15,16,20,21,22,23,24,25,26,27]. Recently, the morphological characterization, geographical distribution, and ethnobotany of many of these species have been described in detail by Ulloa-Inostroza et al. (2017) [15] and Schmeda-Hirschmann et al. (2019) [16]. In this review, we focus on five fruit species growing in Patagonia with high potential as functional food (i.e., maqui, murta, calafate, arrayán, and Chilean strawberry, see Table 1); giving a little background on the fruit quality; and discussing the recent research data available—regarding the particular compound profile, their processing, and clinical assays—and the aspects to consider the commercial prospection of these Patagonian berries.

## 2. Quality Aspects and Bioactive Compounds of Patagonian Berries

### 2.1. Fruit Quality

According to Barrett et al. (2010) [28], in reference to fruits, the characteristics that impart a distinctive quality may be described by four different attributes: color and appearance, flavor (taste and aroma), texture, and nutritional value. All these aspects are determined through the complex biological process of fruit development and ripening [29,30].

We next summarize the main quality aspects of Patagonian berries, such as color and appearance, flavor, and texture. Nutritional and functional value-related antecedents of berries will be addressed in the next sections.

#### 2.1.1. Color and Appearance

The precise definition of the developmental and ripening stages is necessary to determine the physicochemical and physiological parameters that contribute to the different quality attributes of fruit at harvest. A representative fruit at the ripe stage for each species analyzed in this review is shown in Figure 1. In Chilean strawberry, four developmental fruit stages have been described (i.e., small green, C1; large green, C2; turning, C3; and ripe fruit, C4) [31]. The ripe fruit stage (at harvest) has shown a pink receptacle and red achenes that, in comparison with the ripe stage of *Fragaria* x *ananassa* (‘Aromas’) fruit, can be 200-fold less red (comparison of the a* color parameter) [32,33]. Regarding the fruit weight of *F. chiloensis* fruit, it is nearly half of that present in modern commercial strawberry varieties (such *F*. x *ananassa* ‘Chandler’) [31]. In maqui berry, five different maturity stages have been described, starting from 21 days after fruit set in central Chile and named as green (I and II), light red, purple, and dark purple stages [34]. The berry weight per 100 fruits ranges from 10 g (green I stage) to 21 g (dark purple stage), with the highest increase in weight between the green II and light red stages.

The shape of murta and arrayán fruit was reported as globular, with a major equatorial diameter [21,44]. As far as we know, the only report for arrayán fruit development was made by Fuentes et al. (2016) [21]. In that work, the authors classified the fruit development into four stages, mainly by fruit shape and skin color: small and thin green (La1), rounded turning (La2), rounded purple (La3), and black ripe (La4) berries, with a decrease in lightness (L*), b*, and chroma values of fruit skin from approximately 48 to 23 from La1 to La4. As expected, a constant increase in the fresh and dry weight was observed during fruit development, although a higher increment was noted between the La3 and La4 stage [21].

For calafate berry, Arena and Curvetto (2008) [45] described a typical double sigmoid curve for fruit growth, with a constant increase in both the fresh weight and the diameter from 14 to 84 days after full flowering, reaching a maximum of 420 mg and 9.6 mm, respectively.

#### 2.1.2. Flavor

The soluble solids content (SSC) and titratable acidity (TA) are proper predictor parameters for the ripening in several fleshy fruits and are the main determinants for fruit flavor [46,47]. Generally, a reduction of TA and a concomitant increase in SSC are observed during fleshy fruit ripening [48]. Calafate, Chilean strawberry, and arrayán fruits presented this pattern from green to ripe stages [21,49,50,51], but no extensive information has existed in maqui and murta berries until now. The SSC/TA ratio in the arrayán berry increases significantly in the final stages of development [21]. In maqui berry, soluble solids increase during ripening and in dark purple stages, range from 18.8 to 19.9° Brix [34], whereas in the ripe stage of arrayán (black ripe stage), a range between 11.5 to 12.5° Brix was observed [21]. In murta, 22–25% SSC, 4–8 g/L of organic acid (tartaric), and pH 4.7–5.2 were reported in ripe fruit [44]. For the calafate berry, the entire fruit growth period reaches up to 126 days from the full flower, where the fruit presents the highest SSC (25° Brix) and the lowest TA (2.19% to 2.6%) values [45,49]. In this sense, a 4.5-fold increase in the SSC/TA ratio was observed from 56 to 126 days from a full flower in the calafate berry [45]. In calafate, citric and malic acid contents increased during the first stages of fruiting and then decreased toward the end of ripening, although the citric acid content stayed constant from the onset of ripening. Oxalic and tartaric acid contents were maximal between 42 and 70 days from a full flower and then decreased toward the end of the fruiting period [50].

Relatively little information is available regarding the aroma profiles of maqui, calafate, and arrayán berries. In contrast, more detailed information could be found for Chilean strawberry and murta [51,52]. Chilean strawberry fruit is characterized by its great aroma and flavor [51,53]. In this sense, González et al. (2009) [51] identified mainly esters, and secondary alcohols and ketones, with esters and alcohols being up to 73% and 25% of the total volatiles at the ripe stage, respectively. Some esters were reported for the first time in Chilean native strawberry, without references in the commercial strawberry [51], suggesting that the native species has a particular aroma profile. In murta, aroma evolution during storage showed 24 volatile compounds identified, and the concentration of these compounds ranged from 1.2 to 250.5 μg kg^−1^ fresh weight. Methyl 2-methyl butanoate, ethyl butanoate, ethyl hexanoate, ethyl benzoate, ethyl 2-methyl butanoate, methyl hexanoate, and methyl benzoate were the major components, while the most potent compounds in the murtilla fruit aroma were ethyl hexanoate and 4-methoxy-2,5-dimethyl-furan-3-one [52].

#### 2.1.3. Texture

Fruit firmness is one of the leading quality attributes of texture and has an essential commercial impact for both exporters and consumers [54]. In this sense, firmness should be a key goal of breeding in Patagonian soft berries. Strawberry is one of the softest fruits, and loss of firmness is well-documented as being related to cell wall disassembly during ripening [55]. Most significant decreases in cell wall polymers associated with Chilean strawberry fruit ripening occur within the pectin fractions, especially in the covalently bound pectin fraction, which is highly correlated with firmness loss and an increase in the activity of specific cell wall-related enzymes, such as beta-galactosidase [55]. It was reported that the modified atmosphere packaging (MAP) of Chilean strawberry during 12 days of storage at 4 °C delayed the fruit dehydration and the firmness loss, that allow the preservation of quality parameters and anthocyanin compounds compare to fruit storage in the control conditions [56]. However, the application of MAP technology diminished the relative abundance of total volatile compounds [56]. In the arrayán berry, a significant reduction in fruit firmness was observed between rounded purple and black ripe stages [21], although this loss in firmness is slower than that observed during the fruit development of *F. chiloensis* [31]. The firmness reduction of *L. apiculata* fruit [21] showed similar values and trends reported for blueberry fruit [57]. A comparative study of postharvest in the two varieties of murta (i.e., South Pearl INIA and Red Pearl INIA) showed that Red Pearl INIA has a major shelf life during 35 days of storage at 0 °C [58]. The postharvest assay showed a storage capacity of South Pearl INIA during 20 days at 0 °C, while Red Pearl INIA showed major potential for post-harvesting [59]. During treatment of a controlled atmosphere (CA), Red Pearl INIA was stored without problems until 35 days, while South Pearl INIA showed storability until 25 days [59]. 

### 2.2. Antioxidant Capacity

In plants, phenolic compounds are produced as secondary metabolites exerting various protective roles and are generally involved in the defense against stress conditions [60,61,62,63]. The main phenolic compounds in these fruits can be divided into phenolic acids, and flavonoids such as flavonols, flavanols, and anthocyanins (Figure 2) [62,63]. These molecules are responsible for the major organoleptic characteristics of plant food, such as the visual appearance, flavor, bitterness, astringency, and aroma [64]. Many beneficial effects attributed to phenolic compounds [64,65,66,67] have given rise to a new interest in finding plant species with a high phenolic content and relevant biological activity. Studies on the phenolic compounds of the fruits of maqui, murta, calafate, arrayán, and Chilean strawberry highlight the high antioxidant activity they present [15,16,17,18,19,20,21,22,23] (Table 2). In the following section, we briefly summarize the available literature on the main phenolic compounds described for the Patagonian berries analyzed in this review (Table 2).

Different methods have been used for determining the total antioxidants in different vegetables and fruit, including Patagonian berries. Currently, the oxygen-radical absorbing capacity (ORAC) is a method commonly used to compare the antioxidant capacity in different foods [11,73]. The ORAC values (as µmol per 100 g of dry weight, DW) of maqui (37,174), calafate (72,425), murta (43,574), and arrayan (62,500) berries were reported as being higher than in commercial berries such as raspberries, blueberries (*Vaccinium corymbosum* ‘Bluegold’) (27,412), and blackberries cultivated in Chile [11,21,69]. Similar trends were reported using different methods [20]. The Trolox equivalent (TE) antioxidant capacity (TEAC) showed that maqui (88.1) and calafate (74.5) had a higher antioxidant capacity (µmol TE per gram of fresh weight, FW) compared to murta (11.7) and blueberry (14.5) fruits [20]. The analysis by 2,2-diphenylpicrylhydrazyl (DPPH) methods showed that the antioxidant activity (mg of crude extract per liter) was higher in maqui (399.8) than in murta (82.9) [15]. The IC_50_ range of maqui extract (0.0012 and 0.0019 g L^−1^) compared to the average value (0.03 g L^−1^) of commercial berries cultivated in Chile, such as blueberry (*V. corymbosum*), strawberry (*F.* x *ananassa*), and raspberry *(Rubus idaeus*), indicates that a minor concentration of maqui extract is required to inhibit DPPH radicals [74,75]. The above information represents a fundamental background supporting the idea that the Patagonian berries have good potential as a functional food, by themselves or as food ingredients.

#### 2.2.1. Phenolic Content and Composition

The phenolic compounds reported in native Chilean berries include caffeic acid, ferulic acid, gallic acid, myricetin, p-coumaric acid, and others [15,16,17,18,19,20,21,22,23]. Similar to what has been observed for the antioxidant capacity, high total polyphenols contents (TPC) were found for maqui and calafate [19,20]. The different reports of total phenolic analysis using the Folin–Ciocalteu method showed different rankings for Patagonian berries. The first studies showed a higher total phenol content (as μmol gallic acid equivalents (GAE) per gram of FW) for maqui (97 μmol GAE g^−1^ FW) and calafate (87 μmol GAE g^−1^ FW), followed by murta (32 μmol GAE g^−1^ FW) compared to blueberry (17 μmol GAE g^−1^ FW) [20]. Some reports showed similar values of the total polyphenols content (as mg GAE per gram of DW) for calafate (33.9 mg GAE g^−1^ DW), maqui (31.2 mg GAE g^−1^ DW) and murta (34.,9 mg GAE g^−1^ DW) [11], while other reports indicated significant differences between Patagonian berries, with higher values for calafate (65.5 mg GAE g^−1^ DW), followed by arrayán (27.6 mg GAE g^−1^ DW), and lower values for murta (9.2 mg GAE g^−1^ DW) [19].

Concerning the polyphenols composition of the Patagonian berries, maqui and calafate showed anthocyanin as the main component, while fruits of the Myrtaceae family (e.g., murta and arrayán) showed a higher content of flavonoid compounds [15,16,18,19,20,21,70,71,72]. Calafate fruit showed a comparable flavonoid content (0.16 μmol g^−1^ FW) to that obtained for maqui fruit (0.12 μmol g^−1^ FW) [20]. In calafate berry collected from different localities, the identification of flavonoids and phenolic acids showed a higher content of rutin, gallic-chlorogenic, and caffeic acid, and the presence of coumaric and ferulic acid, quercetin, myricetin, and kaempferol [20,76].

The multiple bioactive compounds of the maqui berry (i.e., phenolic antioxidants, alkaloids, flavonoids, and particularly anthocyanins) have contributed to knowledge of the functional potential of this berry in several countries [38,77,78,79]. An HPLC analysis of maqui berry extracts showed 10 compounds identified as flavonols and ellagic acid [70]. The non-anthocyanin compounds were mainly quercetin and its derivatives (with the highest concentration of dimethoxy-quercetin, followed by rutin (quercetin-3-rutinoside) and quercetin-3-galactoside), myricetin and its derivatives, and an important content of ellagic acid [70].

In arrayán, the polyphenol compounds identified mainly correspond to flavonols such as quercetin 3-rutinoside and its derivates, tannins and their monomers, and a minor number of anthocyanins [18,21]. In murta, caffeic acid-3-glucoside, quercetin-3-glucoside, and quercetin were reported as three major compounds in ethanolic extracts of fruit, and the others compounds were gallic acid, rutin, quercitrin, luteolin, luteolin-3-glucoside, kaempferol, kaempferol-3-glucoside, myricetin, and *p*-coumaric acid [72].

In the Chilean strawberry species, several compounds were identified, including an ellagic acid-based compound, catechin, and flavonol derivatives. The higher content of non-anthocyanins identified in *F. chiloensis* and *F.* x *ananassa* ‘Chandler’ were ellagic acid and their pentoside and rhamnoside derivatives and quercetin glucuronide [17]. On the other hand, ellagitannin, quercetin pentoside, and kaempferol glucuronide were only reported in *F. chiloensis* and some compounds—catechin, quercetin pentoside, and quercetin hexoside—were only reported in *Fragaria chiloensis* ssp. *chiloensis* f. chiloensis, and other compounds—procyanidin tetramers and ellagitannin—were only reported in *F. chiloensis* ssp. *chiloensis* f. patagonica [17].

#### 2.2.2. Anthocyanins

Different studies suggest that the highest total anthocyanin content (TAC) can be found in calafate and maqui berries, especially those harvested in the Chilean Patagonia, followed by fruits of the Myrtaceae family species, i.e., arrayán and murta [15,20,80]. It was reported that the total anthocyanin concentrations were higher in calafate fruit extract (between 14 and 26 μmol g^−1^ FW) [20] and (between 23 and 36 μmol g^−1^ FW) [80], followed by maqui berries (between 16 and 20 μmol g^−1^ FW), whereas murta (0.2 μmol g^−1^ FW) showed lowest values than blueberry (2.0 μmol g^−1^ FW) [20].

Similar results were reported by Brito et al. (2014) [19], with a higher anthocyanin content (as mg cyanidin 3-O-glucoside g^−1^ DW) in calafate (51.6), followed by arrayán (15.2) and murta (6.9) berries. The anthocyanin composition of the maqui berry corresponds to 3-glucosides, 3,5-diglucosides, 3-sambubiosides, and 3-sambubioside-5-glucosides of delphinidin and cyanidin, and 34% of total anthocyanins correspond to delphinidin 3-sambubioside-5-glucoside, the major anthocyanin [71,81]. In calafate berry, the main anthocyanins described were delphinidin-3-glucoside, delphinidin-3-rutinoside, delphinidin-3,5-dihexoside, cyanidin-3-glucoside, petunidin-3-glucoside, petunidin-3-rutinoside, petunidin-3,5-dihexoside, malvidin-3-glucoside, and malvidin-3-rutinoside [20]. The above suggests that the antioxidant capacity observed in calafate berries is probably due to their anthocyanin diversity and, in maqui, is due to the particular presence of delphinidin 3-sambubioside-5-glucoside.

Nevertheless, the higher flavonoid content in the Myrtaceae family [11,20], anthocyanins such as peonidin-3-galactoside, petunidin-3-arabinoside, malvidin-3-arabinoside, and peonidin-3-arabinoside, were reported in both the methanol-HCl and methanol extracts of arrayán fruit [21]. The first three have been described in blueberry [82], and delphinidin-3-, malvidin-3-, and peonidin-3-arabinoside; peonidin-3- and malvidin-3-glucoside were described in murta and calafate berries [82]. Other anthocyanins, such as delphinidin-3-arabinoside, cyanidin-3-glucoside, peonidin-3-glucoside, malvidin-3-glucoside, and petunidin-3-arabinoside, were observed in a methanol-HCl extract of arrayán fruit by different authors [21,82].

The two major anthocyanins identified in both Chilean strawberry botanical forms were cyanidin 3-O-glucoside and pelargonidin 3-O-glucoside; these two compounds have generally been described in different *Fragaria* spp. [17]. On the other hand, cyanidin-malonyl-glucoside and pelargonidin-malonyl-glucoside were only reported in Chilean strawberries compared to commercial strawberry (‘Chandler’) [17].

## 3. Effects of Processing on Bioactive Compounds

Many native fruits are only available in determining seasons, so it is difficult to have these fresh fruits for consumption all year or away from collection sites. In general, anthocyanins are susceptible to degradation under environmental conditions, such as oxygen, heat, and changes in pH, among others [83]. The effectiveness, uniformity, and richness of these products are dependent upon the preservation of bioactive compounds throughout the value-added chain. Native berries exhibit high water activity and are highly perishable and susceptible to microbial deterioration, enzymatic reactions, and oxidation [39]. The effects of drying, the microencapsulation process, and juice preparation have been evaluated in maqui and murta berries. In addition, maqui and murta leaf extracts have been evaluated as ingredients to incorporate in food or coating. It was reported that the incorporation of murta leaves extracts in tuna-fish (*Thunnus tynnus*) gelatin-based edible films leads to transparent films with increased protection against UV light and antioxidant capacity [84]. The availability of new products based on maqui and murta as functional ingredients among other Patagonian berries goes hand in hand with the study of the preservation techniques of these fruits. In the following sections, we summarize the literature regarding the effect of processing, with an emphasis on functional maqui and murta products.

### 3.1. Drying Process

Advances in drying technology and standardization techniques in compound analysis allow for the possibility of using drying for the development of functional foods and nutraceuticals. It is essential to consider that the selection of the type of dryer or drying system used for a specific situation must be based upon the product’s characteristics and drying behavior, as well as the end product required [85]. Solar drying (SoD) is the cheapest method for drying whole fruits and vegetables. However, the long drying time and the risk of contamination and spoilage due to exposure to an open environment are the main drawbacks associated with this method. Hot air dryers (HAD) are commonly used by the food industry as they provide relatively fast, uniform, and sanitary drying [86]. However, in most cases, it is possible to modify the richness of bioactive compounds of the raw material, as a function of the temperature/time combination applied in the process [87]. Freeze drying (FD) can produce high-quality products, but is comparatively more expensive; however, and despite this, FD use has been increasing in the industry of processing fruits [88].

Besides its potential role in the battle against certain illnesses and degenerative diseases, some native fruits like maqui or commercial varieties of species like blueberries, cranberries, and tomatoes, share a unique characteristic: a waxy outer skin. The waxy layer affects the flow of moisture from inside the fruit to its surface, which is a crucial process in drying. In the particular case of maqui fruits, the drying process is limited by an external waxy layer similar to that of grapes, which hinders mass water transfer and reduces the drying rate [89]. Technologies and methods applicable to the drying of small waxy skinned fruit could be suitable for obtaining foods and nutraceuticals from maqui fruits. In these cases, several chemical and physical pre-treatments were suggested by several authors to improve the drying rate of whole fruits with waxy skins, e.g., grapes, cherries, plums, apricots, and blueberries [90,91,92,93,94,95]. Pre-treatment methods employing chemical dipping, mechanical processes, and thermal treatments have been used to overcome the wax barrier in several applications [96,97,98,99,100].

Drying technologies are widely used in the industry as a strategy to protect functional molecules—anthocyanins—in value-added products, such as health food ingredients. Convective air drying technologies such as cabinets or trays, fluidized beds, spouted beds, and microwave/spouted beds (MWSB), and those using other technologies (spray-drying, freeze-drying, vacuum, microwave, and osmosis), are some alternatives for processing fruits and vegetables [101]. Between them, spray-drying (SD) is available in the pharmaceutical and food industries [83,102,103,104]. This method is the most used in the food industry because it is economical, rapid, and effective in protecting this compound [105]. For example, SD is widely used in the pharmaceutical and food industries to encapsulate anthocyanin compounds due to the short drying times (5–30 s) [83,102,103,104]. During the last decade, freeze-drying (FD) has become more widespread in the food industry [103]. The FD technique is based on the removal of water from a frozen product by sublimation and has been used as an alternative method to encapsulate anthocyanins [106]. An economically accessible method is vacuum-drying, which allows effective removal of moisture under low pressure, temperature, and oxygen levels, and it is useful for thermolabile products [107].

Regarding the evaluation of the drying process of maqui and murta fruits, it was reported that the preservation techniques—freeze, convective, sun, infrared, and vacuum drying—result in a final maqui product with proper levels of phenolic compounds [38]. All these drying techniques showed a higher content of phenolic and antioxidant compounds, and freeze-dried samples retained over 60% of delphinidin and cyanidin derivatives of fresh fruits [38].

The convective hot air drying (40 to 80 °C) of maqui berries showed that a thermal load and not a high temperature are the main factors that affect the stability of bioactive compounds. At 40 °C, there was a long exposure of the berries to hot air compared to the drying process at 80 °C [39]. Above 60 °C, the bioactive components, such as β-carotene, tocopherols, anthocyanin, and vitamin B6, were not significantly affected, while gallic and ellagic acids increased, as a result of the conversion of hydrolyzable tannins. This phenomenon indicates that the loss of antioxidant activity is compensated for by a probable formation of bioactive components directly related to TPC [39]. Similar studies on murta berries (40 to 80 °C) showed that the β-carotene, total phenolic, and flavonoid contents show a significant decrease during the drying process compared to fresh fruit. However, the ORAC value showed similar antioxidant activity at higher drying temperatures (70–80 °C) compared to fresh fruit [108]. Otherwise, convective and combined convective-infrared conditions at 40, 50, and 60 °C and 400–800 W show that chromaticity coefficients a* and b*, the total surface color difference (ΔE), and TPC are dependent on the mode of heat supply. In addition, a constant temperature and high infrared power 40 °C/800 W reduced the drying time, resulting in dried samples with the highest TPC [42]. A comparative study conducted to evaluate the effect of convective hot-air drying at 65 and 80 °C and freeze drying of bioactive compounds of the Red Pearl-INIA variety of murta fruits showed that freeze-dried fruit retained higher values for TPC (21.924 mg g^−1^ DW) and TAC (0.134 mg g^−1^ DW) than the murta dried by convective hot-air at both temperatures, with a better retention of polyphenols and antioxidant activity during freeze drying [109].

### 3.2. Microencapsulation for Liquid Preparation

The anthocyanin content in maqui is significantly higher compared to other berries, which explains the great interest in its use for nutraceutical purposes. However, these bioactive compounds are highly labile, depending on the stabilization system used [103]. Microencapsulation technology can be used as a strategy to protect maqui anthocyanins in healthy food ingredients. Bastías-Montes et al., (2019) [110], showed that the microencapsulation of maqui can be one way to protect anthocyanins from degradation reactions and can be useful in liquid food preparation, such as for juice and yogurt, with a high content of bioactive compounds. The microencapsulation is a protective technological alternative through which certain bioactive substances in solid, liquid, or gas stage into microparticles with a diameter of 1–1000 μm, and has been widely used in the fields of medicine, cosmetics, food, textile, and advanced materials [111,112,113]. The unique advantage of microencapsulation lies in the fact that the core material is completely coated and isolated from the external environment. The aim is preserving them from various agents, as well as protecting them from oxidation reactions caused by light or oxygen.

Phenolic compounds are phytochemicals extensively metabolized after consumption; thus, the bioavailability should be considered when evaluating the potential health benefits of fruit ingestion. However, bioavailability is influenced by bioaccessibility, which is defined as the relative amount of nutrients or phytochemicals released from a complex food matrix in the lumen of the gastrointestinal tract, becoming available for absorption into the body [114,115]. The comparative analysis of microcapsules of maqui juice powdered by spray-drying or freeze-drying indicated that the morphology and particle size were the most relevant differences and affect the final solubility (70.4–59.5%) in water. However, no significant differences in the stability of anthocyanins in yogurt preparation and in the bioaccessibility after in vitro digestion were observed [104]. Other studies show that the encapsulation with inulin or sodium alginate allows maqui juice spray-drying until 133 days, and the highest encapsulation efficiency of anthocyanins was obtained with inulin. Both maqui juice microparticle methods improved the bioaccessibility (10%) of anthocyanins compared to maqui juice [116]. In murta, comparative studies showed that the highest bioactivity and storability of bioactive phenolics in juice extract were 28 ± 1 min for frozen-thawed fruits and 34 ± 1 min for fresh fruits [117]. In addition, the bioaccessibility index of polyphenols in fresh murta berries or their juice achieved a relatively high value (around 70%) at the end of the small intestine digestive step; however, the juice released the bioaccessible bioactive compounds in the earlier gastric stage, while the fresh fruit increased the release of bioactive compounds in the small intestine [117].

## 4. Healthy Potential of Patagonian berries

Phenolic compounds are effective antioxidants and can display various effects, including anti-microbial, anti-inflammatory, anti-mutagenic, anti-carcinogenic, anti-allergic, anti-platelet, vasodilator, and neuroprotective effects [65,67,118]. These properties have given rise to a new interest in finding plant species with a high phenolic content and relevant biological activity. The epidemiological evidence supporting the benefits of consuming a diet rich in foods containing polyphenols is strong [119,120,121]. In addition to the above, the richness of certain phenolic compounds present in different foods does not guarantee their absorption by the organism, which is how the bioavailability of each of them arises as one of the properties to study to correlate the intake and the effects thereof. The bioavailability appears to differ greatly among the various phenolic compounds, and the most abundant ones in our diet are not necessarily those that have the best bioavailability profile [121,122,123,124]. There has been a broad discussion about whether a high polyphenol content or high antioxidant activity can be associated with a real effect on human health. However, the results related to the preclinical evaluation of the antioxidant capacity and bioactivity of polyphenol extracts using cell cultures, isolated tissues, and animal models, before clinical trials, are still a good approach to understanding the healthy potential of several native fruits. In addition to the advances concerning characterization of the antioxidant capacity and the profile of bioactive molecules in fresh or processed Patagonian berries, advances have been made in the evaluation of the healthy potential of these berries (Figure 3). These sections summarize and discuss the literature regarding the progress in research on the effect of Patagonian fruit extracts in chronic diseases such as metabolic syndrome (MetS), diabetes, and cardiovascular diseases (CVD).

### 4.1. Polyphenols and Anti-Inflammatory Effects

Inflammation is a natural defense mechanism associated with many diseases, such as viral and microbial infections, allergies, obesity, and autoimmune and chronic diseases, and also includes reactions to an unhealthy diet or toxic compounds [120]. During the development of chronic diseases, and due to the higher production of reactive oxygen species (ROS), a series of oxidatives affect various proteins triggering the release of inflammatory signals that can lead to chronic inflammation [120,125]. Anti-inflammatory activity of polyphenols such as quercetin, rutin, morin, hesperetin, and hesperidin has been reported in both acute and chronic inflammation performed in animal models [120]. Polyphenols can exert anti-inflammatory effects by modulating enzymes involved in the metabolism of arginine and arachidonic acid, regulating cell activity, and influencing the production of proinflammatory molecules [120].

The high content of flavonoids, such as quercetin, present in arrayán and murta, suggests its participation as a protective agent in inflammatory diseases. Quercetin (also known as rutin), mainly present as quercetin 3-rutinoside in fruits and vegetables, is a flavonol described in the fruits of calafate, murta, and arrayán; a high concentration of quercetin in the methanolic extract obtained from the arrayán fruit has been observed [15,21]. Purified quercetin has a variety of biological effects, including antiallergic, anti-inflammatory, antioxidant, and platelet antiaggregant effects [126]. In addition, potential protective effects against acute lung injury (ALI) induced by endotoxin or lipopolysaccharide (LPS), a component of the cell wall of Gram-negative bacteria, have been described [127]. In mice, the previous administration of quercetin inhibits several mechanisms associated with the inflammatory process during pulmonary infection, such as the inhibition of arterial blood gas exchange induced by LPS and the infiltration of neutrophils in the lungs, suppression of LPS-induced expression of the macrophage inflammatory protein (MIP)-2, inactivation of matrix metalloproteinase (MMP)-9, and inhibition of Akt phosphorylation [127].

Studies conducted in animal models suggest that polyphenols in the diet have a positive effect on lung injury [128]. The inhalation of quercetin in radiation-induced pneumonitis in rats increases the number of leukocytes and erythrocytes in the blood, reduces the number of inflammatory cells in the bronchoalveolar lavage fluid, reduces hemorrhage and the infiltration of inflammatory cells, and suppresses the expression of proinflammatory cytokines that transform the growth factor β1 and interleukin-6 [128], suggesting that the inhalation of flavonols has the potential to become a new alternative in the treatment of lung diseases such as radiation pneumonitis.

As we previously stated, the maqui berry is the richest known natural source of delphinidins. An in vitro assay of this purified molecule showed an increase in the generation of nitrogen oxide (NO) in endothelial cells, decreased platelet adhesion, and anti-inflammatory effects. Additionally, it has been reported that delphinidins can counteract aging of the skin and inhibit osteoporosis [26]. Aqueous extracts of maqui berry prevent the oxidation of low-density lipoproteins (LDL) induced by copper, protect the cultures of human endothelial cells, and have anti-adipogenic and anti-inflammatory effects [75,78,129,130]. The extracts of maqui and calafate fruits have inhibitory properties of the inflammatory response generated by the interaction of adipocytes and macrophages [27]. These extracts showed a reduction of nitric oxide (NO) production, inhibition of the induction of nitric oxide synthase (NOS) and TNF-alpha, and induction of the interleukin 10 (IL-10) gene expression; on this basis, it has been suggested that they could be potential therapeutic tools against the comorbidity associated with the development of obesity [27].

An in vitro assay performed in LPS-activated murine macrophage RAW-264 cells showed that extracts and subfractions of maqui berry, and also quercetin, gallic acid, luteolin, and myricetin, suppressed the LPS-induced production of NO, by downregulating iNOS and COX-2 expressions; according to the authors, the phenolic compounds anthocyanins, flavonoids, and organic acids, as the fractions, may provide a potential therapeutic tool for inflammation-associated disorders [131]. The antioxidant and anti-inflammatory effects of water extracts of maqui berry were tested in a mouse dermatitis model showing an increase of interferon-gamma (IFN-γ) levels and a decrease of interleukin-4 (IL-4), suggesting its potential use for atopic dermatitis treatment [132].

Studies in humans showed that anthocyanin maqui extract normalized H_2_O_2_ and IL-6 concentrations in exhaled breath condensates (EBC) by asymptomatic smokers [133], suggesting that the maqui could be considered an interesting alternative for dietary management in patients with respiratory disorders. Another study showed that the extracts of leaves and berries of murta have a strong anti-inflammatory activity when applied topically in mice, due to several pentacyclic triterpene acids, including the 2-a-hydroxy derivatives alphitolic, asiatic, and corosolic acids [129,134,135,136].

#### 4.1.1. Polyphenols and Metabolic Syndrome

Metabolic syndrome (MetS) includes several metabolic abnormalities, such as abdominal obesity, hypertension, insulin resistance, and dyslipidemia. MetS has been associated with an increased risk of CVD and type 2 diabetes mellitus (T2DM) [67]. The onset and progression of MetS are mediated by body weight and blood pressure reduction, as well as improvement in insulin-sensitivity and lipid metabolism [119]. The beneficial effects of polyphenols, mainly flavonoids, are associated with their interaction with several molecular pathways involved in the metabolism of glucose and the regulation of insulin-signaling pathways [67].

A remarkable activity of polyphenols is their ability to retard carbohydrate digestion by the direct inhibition of enzymes, such as R-glucosidase and R-amylase [137]. As a result, the inhibition of these enzymes reduces the glucose absorption rate. It was reported that the crude extract of murta and maqui leaf rich in polyphenols—lavan-3-ol polymers, quercetin glucoside, and kaempferol glucoside—showed an effective inhibitory effect by a non-competitive mechanism on R-amylase and R-glucosidase f [137]. The above suggest a potential effect of these extracts in regulating postprandial hyperglycemia. In a murine model of type II diabetes, the oral administration of a standardized anthocyanin-rich formulation from maqui and pure delphinidin 3-sambubioside-5-glucoside (D3S5G) showed a dose-dependent decrease of fasting blood glucose levels and glucose production in rat liver cells [138].

A clinical trial conducted on individuals with a moderate glucose intolerance, daily supplemented with 180 mg Delphinol®, a standardized, water-soluble maqui berry extract, for three months, showed a progressive decrease of glycosylated hemoglobin, reduction of LDL and VLDL after one month, and increase of HDL from the baseline during the entire treatment period, without changes of total cholesterol and triglycerides, suggesting that longer treatment has a better effect on the glycemic and lipid profile [24,25]. A clinical pharmacokinetic study showed that after single-dose supplementation with Delphinol®, delphinidin-3-*O*-glucoside, and cyanidin-3-*O*-sambubioside, the selected anthocyanins in the assay reached the maximal concentration after approximately 1 and 2 h, respectively, confirming the bioavailability of these anthocyanins, and also their fast uptake and metabolism [139].

#### 4.1.2. Cardiovascular Effects

CVD is the primary cause of mortality and morbidity worldwide. There is substantial evidence that early events of asymptomatic hyperglycemia increase the risk of CVD, even in the absence of diabetes [140]. Hyperglycemia is associated with endothelial dysfunction, characterized by reduced endothelium-dependent vasodilation (EDV), which is usually used as a measure to prove endothelial function in different pathological conditions [141].

Murta and arrayán berries might have beneficial effects on the management of cardiovascular diseases. The vasoprotective activity of the extract of these fruits could be associated with a cocktail of different molecules rather than a particular molecule. It was reported that a murta extract rich in gallic acid, catechin, quercetin-3-β-D-glucoside, myricetin, quercetin, and kaempferol did not generate toxic effects on human endothelial cells and had significant antioxidant activity against lipid peroxidation and superoxide anion and ROS production [142]. The same extract showed a dose-dependent vasodilator activity in aortic rings in the presence of endothelium, whose hypotensive mechanism is partially mediated by large conductance calcium-dependent potassium channels and nitric oxide synthase/guanylate cyclase [142]. Conversely, a methanolic extract of arrayán fruit harvested from a natural population located at Antuco (Biobio Region, Chile) containing quercetin-3-rutinoside, petunidin-3-arabinoside, peonidin-3-galactoside, malvidin-3-arabinoside, and peonidin-3-arabinoside arabinoside, showed vasoprotection properties [21]. Briefly, the methanolic extract of arrayán fruit showed dose-dependent (0.1, 1, and 10 mg/mL) protection of the acetylcholine-induced relaxation carried out in rat aortic rings (isolated from same litter animals) preincubated with a high level of glucose, a condition that drastically affects the endothelium-dependent relaxation induced by acetylcholine [21]. The above results suggest that the extract of Patagonian berries may act as a vasoprotector, which allows them to be projected as useful tools to prevent and treat diseases associated with vascular damage induced by high glucose levels (e.g., postprandial hyperglycemia) [143].

Patagonian fruits not only have a high content of polyphenolic compounds, but also have other vasoactive compounds. It was reported that the alkaloid 8-oxo-9-dihydromakomakine extracted from maqui leaves induced a dose-dependent relaxation of aortic rings precontracted with phenylephrine; the induced vasorelaxation was independent of endothelium and partially reduced plasma membrane depolarization-induced contraction, suggesting a protective effect of maqui alkaloids in the treatment of cardiovascular pathologies [144].

A clinical trial conducted in healthy, overweight, and smoker subjects showed that the daily consumption of anthocyanins was associated with reduced levels of oxidative damage markers in plasma (oxidized low-density lipoprotein; Ox-LDL) and urine (F2-isoprostanes). The values returned to the baseline value after 40 days of follow-up, and no significant differences were observed for anthropometric characteristics, ambulatory blood pressure, and the lipid profile [145].

## 5. Some Commercial Aspects

In Chile, maqui and murta are the primary Patagonian berries marketed, and most of them are exported for consumers worldwide (Figure 3). Concerning maqui, the principal harvest is from woodland shrubs. Although, according to the Center of Native plants of Chile (Universidad de Talca), this university published the applications of three domesticated varieties of maqui for their commercial use in the Official Gazette of Chile [146]. Romo and Bastías, (2016) [40] reported that, in 2016, there were 21 companies in Chile related to maqui commercialization since 2009. The Chilean market is focused on the preparation of beverages or juices based on maqui berry. Of these, 13 companies are located in the Metropolitan Region (62%), and the rest is distributed in the other regions, concentrated between the Maule Region and that of Araucanía [40]. In turn, maqui berry-based products can be found in the international market as frozen, juiced, dehydrated, canned, and other fruit preparations. During 2018, the maqui production in Chile included (i) 79,132 Kg of frozen fruit with a worth of US $ 598,207 and a mean value of 7.6 US/kg; and (ii) 3,870 Kg of drying fruit with a worth of US $ 105,269 and a mean value of 27.2 US/kg [147]. The main target markets were the USA, South Korea, Germany, and Japan. According to the Forest Institute (INFOR), 75% of maqui berries are exported freeze-dried [148].

In Chile, murta harvest is from woodland shrubs and domesticate varieties [41]. In 1996, the Agricultural Research Institute of Chile (INIA) developed a domestication program that began with the collection of wild germplasm [15,136]. This program included the development of protocols of plant multiplication [149,150], and the study of genetic diversity by molecular, phenotypic, and agronomic characterization of the wild germplasm [150]. According to a prospective study for new food introduction in the European Union requested by the Chilean Office of Agricultural Studies and Policies (OPEPA) during 2016, the exportation of principally fresh fruit was close to 3,000 Kg, with a worth of US $80,000. The major exportation markets were Italy, Korea, and France, among others [41]. No available information about the arrayán commercialization or breading program was found. However, some companies are interested in including some functional arrayán derivate products.

According to the novel food catalog of the European Union, maqui berry has an authorized use only as or in food supplements, and any other food uses have to be approved for the EU-Novel Food Regulation [151]. Regarding murta berry, the information currently available suggests that this fruit meets the requirements for the novel food solicitation [41].

Concerning Chilean native strawberry, no agro-industrial products have been generated, and this could be because production volumes are low enough to satisfy the demand for raw materials [43] (Figure 3). However, the “Slow Food” Foundation for Biodiversity, which promotes the protection of the biodiversity of food and its environmentally friendly production around the world, has incorporated the Chilean strawberry in the world project known as “The Ark of Taste” (Slow Food Foundation for Biodiversity, 2015) [152]. This critical tendency, associated with the rescue of gastronomic traditions and the growing market gourmet in Chile, can contribute to generating Chile’s public policies regarding protection of the cultural and gastronomic heritage.

With regard to calafate, INIA coordinated the grant conducted for the generation of new varieties for a natural color generation. The project “Territorial Pole for the development of high value colorants and antioxidants for the food industry from highly dedicated raw materials produced in the south-central zone of Chile” includes the participation of INIA and agro-industrial companies and it is an initiative of “Territorial Poles of Strategic Development” created by the Foundation for Agrarian Innovation (FIA), with resources provided by the Strategic Investment Fund (FIE) [153].

## 6. Conclusions

This review provides relevant information about the native Patagonian berries—maqui, murta, calafate, arrayán, and Chilean strawberry—that could be used as a functional food due to its diverse and high flavonol and anthocyanin contents that can prevent inflammatory-, metabolic syndrome-, and cardiovascular-associated pathologies. Within the fruits discussed in this review, maqui is the native berry with major potential, followed by the murta fruit. Future functional studies and the production of cultivars are critical to strengthening the potential of these fruits in the food market.

## Figures and Tables

**Figure 1 foods-08-00289-f001:**
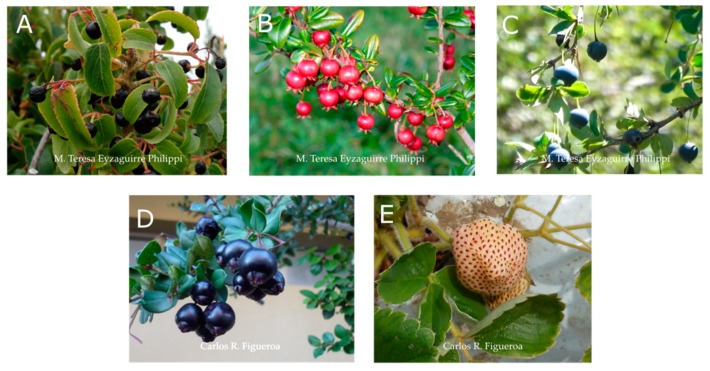
Patagonian berries with healthy potential as a functional food on the basis of recent research data available. (**A**) *Aristotelia chilensis* (Mol.) Stuntz (maqui)*; (**B**) *Ugni molinae* Turcz. (murta)*; (**C**) *Berberis microphylla* G. Forst. (calafate)*; (**D**) *Luma apiculata* (DC.) Burret (arrayan)**; (**E**) *Fragaria chiloensis* (L.) Mill. (Chilean strawberry)**. Photography credit to M. Teresa Eyzaguirre-Philippi (*) and Carlos R. Figueroa (**).

**Figure 2 foods-08-00289-f002:**
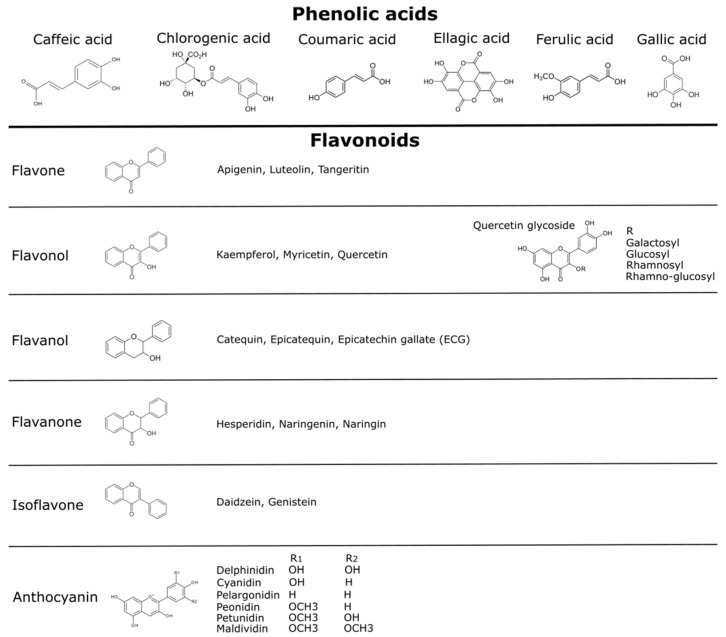
Polyphenols compounds described in vegetables and fruits. Different phenolic compounds have been reported in native Chilean berries, including phenolic acid, flavonoids such as quercetins—principally quercetin glycosides—and anthocyanins [15,16,17,18,19,20,21,22,23]. More details are presented in the text. Chemical structures credits [68].

**Figure 3 foods-08-00289-f003:**
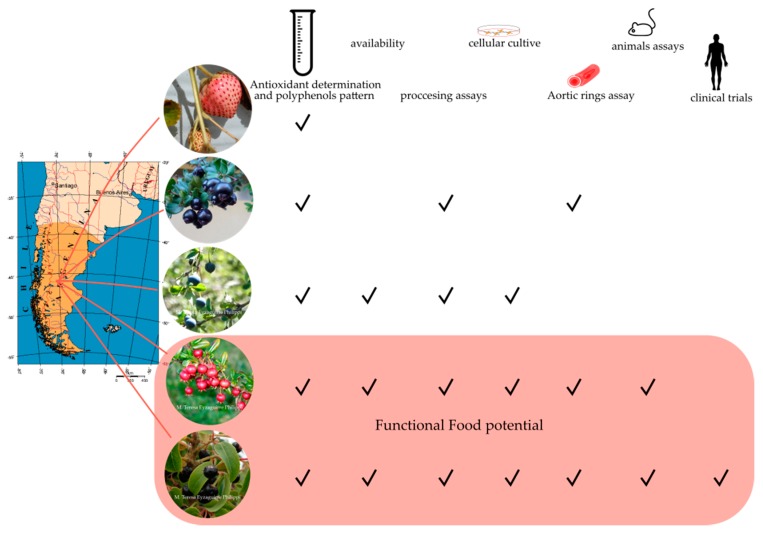
Summary of the Patagonian berries path to becoming functional foods. Maqui* is the native berry of Chile with major research progress concerning processing and the effect on chronic diseases. Murta* is the second most studied native berry, and two domesticated varieties are available in the market. Future studies are critical to strengthening the potential of arrayán**, calafate*, and Chilean strawberry** fruits. More details in the text. Photography credit to M. Teresa Eyzaguirre-Philippi (*) and Carlos R. Figueroa (**), map figure credit to commons.wikimedia.org/wiki/File:Pat_map.PNG, tube figure credit to https://thenounproject.com/term/test-tube/5544/, mouse figure credit to https://www.svgrepo.com/svg/53826/mouse, human figure credit to https://www.flaticon.com/free-icon/standing-human-body-silhouette_30473.

**Table 1 foods-08-00289-t001:** Main features of Patagonian fruits analyzed in the present review. Scientific and common names, botanic family, geographic distribution, traditional products and uses, and functional products generated in the last years.

Species	Common Name	Family	Geographic Distribution [16,35]	Traditional Products and Uses	Functional Products
*Aristotelia chilensis* (Mol.) Stuntz.	Maqui	Elaeocarpaceae	Chile: from the Coquimbo to Aysén regions, including Juan Fernández Island (Latitude 31°–40°). Argentina: from Jujuy to Chubut provinces.	Fresh and dried fruit, use to make textile pigment, cake, jam, juice, alcoholic beverages [36,37]	Freeze-dried maqui (powder and capsules), honey mix, functional drinks, drugs [24,25,26,38,39,40,41]
*Ugni molinae* Turcz.	Murta	Myrtaceae	Chile: From the O’Higgins to Aysén regions, including Juan Fernández Island (Lat. 34°–40°). Argentina: Neuquén, Rio Negro, and Chubut provinces.	Fresh and dried fruit, textile pigment, bakery, jam, alcoholic beverages [37]	Freeze-dried murta (powder and capsules), honey mix [41,42]
*Berberis microphylla* G. Forst.	Calafate	Berberidaceae	Chile: From the Metropolitan to Magallanes regions (Lat. 33°–55°). Argentina: From Neuquén to Tierra del Fuego provinces.	Fresh fruit, used to make jam, juice, beer [36,37]	Natural colorants [37]
*Luma apiculata* (DC.) Burret.	Arrayán	Myrtaceae	Chile: From the Coquimbo to Aysén regions (Lat. 31°–40°). Argentina: From Neuquén to Chubut provinces.	Fresh fruit, textile pigment, bakery, jam, aromatic wine [22,23]	N.D.
*Fragaria chiloensis* (L.) Mill.	Chilean strawberry	Rosaceae	Chile: From the O’Higgins to Magallanes regions (Lat. 34°–55°). Argentina: Neuquén and Rio Negro provinces.	Fresh fruit, used to make alcoholic beverages, cake [36,43]	N.D.

Geographic distribution according to Rodriguez et al., 2018 [35] and Schmeda et al., 2019 [16]. N.D.: not described.

**Table 2 foods-08-00289-t002:** Antioxidant information of Patagonian berries.

Species Name	Average Antioxidant Capacity Determined by ORAC (µmol·100 g DW^−1^) ^a^	Average Range of Total Polyphenols Compounds Content (mg GAE g^−1^ DW^−1^) ^a^	Number of Non-Anthocyanin Polyphenol Compounds Reported	Principal Non-Anthocyanin Polyphenol Compounds	Number of Anthocyanin Compound Reported	Principal Anthocyanin Compounds
Maqui.	37,174 [11,69]	49.7 [70]	13 [15]	Quercetin, dimethoxy-quercetin, quercetin-3-rutinoside, quercetin-3-galactoside, myricetin and its derivatives (dimethoxy-quercetin) and ellagic acid [70]	8 [15]	3-glucosides, 3,5-diglucosides, 3-sambubiosides and 3-sambubioside-5-glucosides of cyanidin and delphinidin (delphinidin 3-sambubioside-5-glucoside) [20,71]
Murta	43,574 [11,69]	9.2 [19] 34.9 [69]	16 [15]	caffeic acid-3-glucoside, quercetin-3-glucoside, quercetin, gallic acid, quercetin-3-rutinoside, quercitrin, luteolin, luteolin-3-glucoside, kaempferol, kaempferol-3-glucoside, myricetin and p-coumaric acid [72]	11 [15]	delphinidin-3-, malvidin-3- and peonidin-3-arabinoside; peonidin-3- and malvidin-3-glucoside [20,72]
Calafate	72,425 [11,69]	33.9 [69] 65.5 [19]	36 [15]	quercetin-3-rutinoside, gallic- and chlorogenic acid, caffeic and the presence of coumaric- and ferulic acid, quercetin, myricetin, and kaempferol [19]	30 [15]	delphinidin-3-glucoside, delphinidin-3-rutinoside, delphinidin-3,5-dihexoside, cyanidin-3-glucoside, petunidin-3-glucoside, petunidin-3-rutinoside, petunidin-3,5-dihexoside, malvidin-3-glucoside and malvidin-3-rutinoside [19,20]
Arrayán	62,500 [21]	27.6 [19]	13 [15]	quercetin 3-rutinoside and their derivatives, tannins and their monomers [18,21]	8 [15]	peonidin-3-galactoside, petunidin-3-arabinoside, malvidin-3-arabinoside, peonidin-3-arabinoside delphinidin-3-arabinoside, cyanidin-3-glucoside, peonidin-3-glucoside and malvidin-3-glucoside [18,19,21]
Chilean strawberry	N.R.	N.R	16*20** [17]	ellagic acid and their pentoside- and rhamnoside derivatives. quercetin glucuronide, ellagitannin, quercetin pentoside, kaempferol glucuronide. Catechin *, quercetin pentosid *, and quercetin hexoside * procyanidin tetramers ** and ellagitannin ** [17]	4 [17]	cyanidin 3-*O*-glucoside, pelargonidin 3-*O*-glucoside, cyanidin-malonyl-glucoside and pelargonidin-malonyl- glucoside [17]

The table shows the available data concerning the antioxidant capacity determined by oxygen-radical absorbing capacity (ORAC) (µmol·100 gDW^−1^), total polyphenols compounds content (mg GAE gDW^−1^), and polyphenol compounds reported in these fruits. N.R.: not reported. (*) polyphenols compounds reported in *F. chiloensis* ssp. *chiloensis* f. chiloensis and reported in (**) *Fragaria chiloensis* ssp. *chiloensis* f. patagonica. More details are given in the text. ^a^ DW, dry weight; GAE, gallic acid equivalents.

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
