# Peer review of "Patagonian Berries: Healthy Potential and the Path to Becoming Functional Foods"

_foods, 2019, doi:10.3390/foods8080289_

Round 1
Reviewer 1 Report
This review provides relevant information about the native Patagonian fruits.
The title "Patagonian berries: healthy potential and the path to becoming functional foods" describes the review.
Along the text where is et al. should be et al. [italic] from the latin “et alia”, meaning “and others.”
The Keywords should be reduced; one suggestion is: Maqui; Murta; Calafate; Arrayan; Chilean strawberry; Berries; Functional foods
Author Response
Response to specific comments of Reviewer 1:
Point 1: Along the text where is et al. should be et al. [italic] from the latin “et alia”, meaning “and others.”
Point 2: The Keywords should be reduced; one suggestion is: Maqui; Murta; Calafate; Arrayan; Chilean strawberry; Berries; Functional foods
As Reviewer 1 suggested, all these changes were done.
The authors really thank the Reviewer 1 comments
Reviewer 2 Report
In Table 1 please include references
In 2.1. "Food quality could be defined as the composite of those attributes that have a significance in 82 determining the degree of acceptability to the consumer" and the next sentence "According to Barrett et al. (2010) [27] ..." express different ideas. According to scientific bibliography I suggest delate the first sentence.
Review citation system, f.e. in L211
Expressions such as "super-fruit" should be avoided.
Some structures may be included in each group.
L394/395 "the evidence for a specific fruit or vegetable and specific phenolic compounds is less convincing and generates controversy" More than one cite could be found.
Some studies should be better explained f.i. L520, How was achieved the extract? Dose of extract administered? Different groups of rats?
Conclusions should be more concrete.
Author Response
Response to specific comments of Reviewer 2:
Point 1: In Table 1 please include references
As reviewer 2 suggests, references were included in Table 1
Point 2: In 2.1. "Food quality could be defined as the composite of those attributes that have a significance in 82 determining the degree of acceptability to the consumer" and the next sentence "According to Barrett et al. (2010) [27] ..." express different ideas. According to scientific bibliography I suggest delate the first sentence.
The first sentence was deleted.
Point 3: Review citation system, f.e. in L211
Point 4: Expressions such as "super-fruit" should be avoided.
Both suggestions were done.
Point 5: Some structures may be included in each group.
As reviewer 2 suggests, structures of compounds were included in a new figure 2.
Point 6: L394/395 "the evidence for a specific fruit or vegetable and specific phenolic compounds is less convincing and generates controversy" More than one cite could be found.
As reviewer 2 suggests, the sentence was improved and more cites were included.
Point 7: Some studies should be better explained f.i. L520, How was achieved the extract? Dose of extract administered? Different groups of rats?
As reviewer 2 suggests, more details of this study were included in the text.
Four aortic rings were obtained from each animal, there were 2 controls and 2 high glucose assays. The experiment is repeated with 4 animals from the same litter (this is a requirement of the trials with aortic ring). Furthermore, the parameters obtained are the product of a non-linear regression so that they have a very low dispersion. In this specific case, the differences are very significant. More details in the reference Fuentes et al., 2016
Point 8: Conclusions should be more concrete.
As reviewer 2 suggests, the conclusion was improved.
The authors really thank the Reviewer 2 comments